# GPT4RoI: Instruction Tuning Large Language Model on Region-of-Interest

## Abstract

Visual instruction tuning large language model (LLM) on image-text pairs has achieved general-purpose vision-language abilities. However, the lack of region-text pairs limits their advancements to fine-grained multimodal understanding. In this paper, we propose *spatial instruction tuning*, which introduces the reference to the region-of-interest (RoI) in the instruction. Before sending to LLM, the reference is replaced by RoI features and interleaved with language embeddings as a sequence. Our model GPT4RoI, trained on 7 region-text pair datasets, brings an unprecedented interactive and conversational experience compared to previous image-level models. (1) *Interaction beyond language*: Users can interact with our model by both language and drawing bounding boxes to flexibly adjust the referring granularity. (2) *Versatile multimodal abilities*: A variety of attribute information within each RoI can be mined by GPT4RoI, *e.g.*, color, shape, material, action, *etc*. Furthermore, it can reason about multiple RoIs based on common sense. On the Visual Commonsense Reasoning (VCR) dataset, GPT4RoI achieves a remarkable accuracy of 81.6%, surpassing all existing models by a significant margin (the second place is 75.6%) and almost reaching human-level performance of 85.0%. The code, dataset, and demo can be found at `https://github.com/Anonymous-Researcher1/GPT4RoI`.

## 1 Introduction

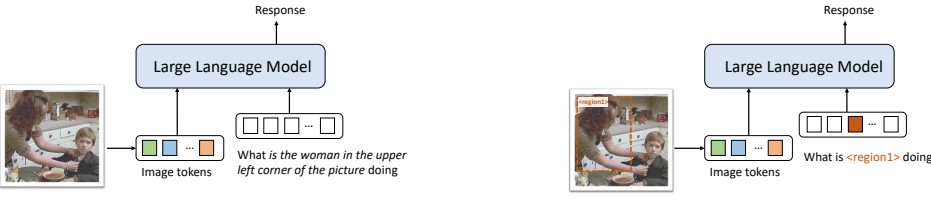

Figure 1: Comparison of visual instruction tuning on image-text pairs and spatial instruction tuning on region-text pairs. The bounding box and text description of each object are provided in region-text datasets. During training, the bounding box is from annotations, and in inference, it can be provided by user or any off-the-shelf object detector

Recent advancements of large language models (LLM) have shown incredible performance in solving natural language processing tasks in a human-like conversational manner, for example, commercial products (OpenAI, 2022; Anthropic, 2023; Google, 2023; OpenAI, 2023) and community open-source projects (Touvron et al., 2023a;b; Taori et al., 2023; Chiang et al., 2023; Du et al., 2022; Sun & Xipeng, 2022). Their unprecedented capabilities present a promising path toward general-purpose artificial intelligence models. Witnessing the power of LLM, the field of multimodal models (Yang et al., 2023c; Huang et al., 2023; Girdhar et al., 2023; Driess et al., 2023) is developing a new technology direction to leverage LLM as the universal interface to build general-purpose models,

| Model | Image | Region | Multi-Region | Multi-Round Dialogue | End-to-End Model |
|---|:---:|:---:|:---:|:---:|:---:|
| Visual ChatGPT (Wu et al., 2023) | ✓ | ✗ | ✗ | ✓ | ✗ |
| MiniGPT-4 (Zhu et al., 2023) | ✓ | ✗ | ✗ | ✓ | ✓ |
| LLaVA (Liu et al., 2023a) | ✓ | ✗ | ✗ | ✓ | ✓ |
| InstructBLIP (Dai et al., 2023) | ✓ | ✗ | ✗ | ✓ | ✓ |
| MM-REACT (Yang et al., 2023c) | ✓ | ✓ | ✓ | ✓ | ✗ |
| InternGPT (Liu et al., 2023d) | ✓ | ✓ | ✓ | ✓ | ✗ |
| VisionLLM (Wang et al., 2023b) | ✓ | ✓ | ✗ | ✗ | ✓ |
| CaptionAnything (Wang et al., 2023a) | ✓ | ✗ | ✗ | ✗ | ✗ |
| DetGPT (Pi et al., 2023) | ✓ | ✓ | ✗ | ✓ | ✗ |
| GPT4RoI | ✓ | ✓ | ✓ | ✓ | ✓ |

Table 1: Comparisons of vision-language models. Our GPT4RoI is an end-to-end model that supports region-level understanding and multi-round conversation.

where the feature space of a specific task is tuned to be aligned with the feature space of pre-trained language models.

As one of the representative tasks, vision-and-language models align the vision encoder feature to LLM by instruction tuning on image-text pairs, such as MiniGPT-4 (Zhu et al., 2023), LLaVA (Liu et al., 2023a), InstructBLIP (Dai et al., 2023), etc. Although these works achieve amazing multimodal abilities, their alignments are only on image-text pairs (Chen et al., 2015; Sharma et al., 2018; Changpinyo et al., 2021; Ordonez et al., 2011; Schuhmann et al., 2021), the lack of region-level alignment limits their advancements to more fine-grained understanding tasks such as region caption (Krishna et al., 2017) and reasoning (Zellers et al., 2019a). To enable region-level understanding in vision-language models, some works attempt to leverage external vision models, for example, MM-REACT (Yang et al., 2023c), InternGPT (Liu et al., 2023d) and DetGPT (Pi et al., 2023), as shown in Table 1. However, their non-end-to-end architecture is a sub-optimal choice for general-purpose multi-modal models.

Considering the limitations of previous works, our objective is to construct an end-to-end vision-language model that supports fine-grained understanding on region-of-interest. Since there is no operation that can refer to specific regions in current image-level vision-language models (Zhu et al., 2023; Liu et al., 2023a; Zhang et al., 2023c; Dai et al., 2023), our key design is to incorporate references to bounding boxes into language instructions, thereby upgrading them to the format of *spatial instructions*. For example, as shown in Figure 1, when the question is *"what is <region1> doing?"*, where the *<region1>* refers to a specific region-of-interest, the model will substitute the embedding of *<region1>* with the region feature extracted by the corresponding bounding box. The region feature extractor can be flexibly implemented by RoIAlign (He et al., 2017) or Deformable attention (Zhu et al., 2020).

To establish fine-grained alignment between vision and language, we involve region-text datasets in our training, where the bounding box and the text description of each region are provided. The datasets are consolidated from publicly available ones including COCO object detection (Lin et al., 2014), RefCOCO (Yu et al., 2016), RefCOCO+ (Yu et al., 2016), RefCOCOg (Mao et al., 2016), Flickr30K entities (Plummer et al., 2015), Visual Genome(VG) (Krishna et al., 2017) and Visual Commonsense Reasoning(VCR) (Zellers et al., 2019a). These datasets are transformed into spatial instruction tuning format. Moreover, we incorporate the LLaVA150K dataset (Liu et al., 2023a) into our training process by utilizing an off-the-shelf detector to generate bounding boxes. This enhances our model's ability to engage in multi-round conversations and generate more human-like responses.

The collected datasets are categorized into two types based on the complexity of the text. First, the plain-text data contains object category and simple attribute information. It is used for pre-training the region feature extractor without impacting the LLM. Second, the complex-text data often contains complex concepts or requires common sense reasoning. We conduct end-to-end fine-tuning of the region feature extractor and LLM for these data.

Benefiting from spatial instruction tuning, our model brings a new interactive experience, where the user can express the question to the model with language and the reference to the region-of-interest. This leads to new capacities beyond image-level understanding, such as region caption and complex region reasoning. As a generalist, our model GPT4RoI also shows its strong region understanding ability on three popular benchmarks, including the region caption task on Visual Genome (Krishna et al., 2017), the region reasoning task on Visual-7W (Zhu et al., 2016) and Visual Commonsense Reasoning (Zellers et al., 2019a) (VCR). Especially noteworthy is the performance on the most challenging VCR dataset, where GPT4RoI achieves an impressive accuracy of 81.6%, 6 points ahead of the second-place and nearing the human-level performance benchmarked at 85.0%.

In summary, our work makes the following contributions:

- We introduce spatial instruction, combining language and the reference to region-of-interest into an interleave sequence, enabling accurate region referring and enhancing user interaction.

- By spatial instruction tuning LLM with massive region-text datasets, our model can follow user instructions to solve diverse region understanding tasks, such as region caption and reasoning.

- Our method, as a generalist, outperforms the previous state-of-the-art approach on a wide range of region understanding benchmarks.

## 2 RELATED WORK

### 2.1 LARGE LANGUAGE MODEL

The field of natural language processing (NLP) has achieved significant development by the high-capability large language model (LLM). The potential of LLM is first demonstrated by pioneering works such as BERT (Devlin et al., 2018) and GPT (Radford et al., 2018). Then scaling up progress is started and leads to a series of excellent works, for example, T5 (Raffel et al., 2020), GPT-3 (Brown et al., 2020), Flan-T5 (Chung et al., 2022), PaLM (Chowdhery et al., 2022), etc. With the growth of training data and model parameters, this scaling up progress brings to a phenomenal product, ChatGPT (OpenAI, 2022). By generative pre-trained LLM and instruction tuning (Ouyang et al., 2022) on human feedback, ChatGPT shows unprecedented performance on conversations with humans, reasoning and planning tasks (Mu et al., 2023; Yang et al., 2023a; Bubeck et al., 2023), etc.

### 2.2 LARGE VISION-LANGUAGE MODEL

To utilize high-performance LLM to build up vision-language models, LLM as task coordinator is proposed. Given the user instruction, LLM parses the instruction and calls various external vision models. Some representative works are Visual ChatGPT (Wu et al., 2023), ViperGPT (Surís et al., 2023), MM-REACT (Yang et al., 2023c), InternGPT (Liu et al., 2023d), VideoChat (Li et al., 2023b), etc. Although these models largely expand the scope of multimodal models, they depend on external vision models and these non-end-to-end architectures are not the optimal choice for multi-modal models. To obtain end-to-end vision-language models, instruction tuning LLM on image-text pairs is proposed to align visual features with LLM and accomplish multimodal tasks in a unified way, for example, Flamingo (Alayrac et al., 2022), MiniGPT-4 (Zhu et al., 2023), LLaVA (Liu et al., 2023a), LLaMa-Adapter (Zhang et al., 2023c), InstructBLIP (Dai et al., 2023), MM-GPT (Gong et al., 2023), VPGTrans (Zhang et al., 2023a), etc. These models achieve amazing image-level multimodal abilities, while several benchmarks such as LVLM-eHub (Xu et al., 2023) and MMBench (Liu et al., 2023c) find that these models still have performance bottlenecks when need to be under specific region reference. Our GPT4RoI follows the research line of visual instruction tuning and moves forward region-level multimodal understanding tasks such as region caption (Krishna et al., 2017) and reasoning (Zellers et al., 2019a).

### 2.3 REGION-LEVEL IMAGE UNDERSTANDING

For region-level understanding, it is a common practice in computer vision to identify potential regions of interest first and then do the understanding. Object detection (Ren et al., 2015; Carion et al., 2020; Zhu et al., 2020; Zang et al., 2023) tackles the search for potential regions, which are generally accompanied by a simple classification task to understand the region's content. To expand

the object categories, (Kamath et al., 2021; Liu et al., 2023b; Zhou et al., 2022; Li* et al., 2022) learn from natural language and achieve amazing open-vocabulary object recognition performance. Region captioning (Johnson et al., 2015; Yang et al., 2017; Wu et al., 2022) provides more descriptive language descriptions in a generative way. Scene graph generation (Li et al., 2017; Tang et al., 2018; Yang et al., 2022) analyzes the relationships between regions by the graph. The VCR (Zellers et al., 2019b) dataset presents many region-level reasoning cases and (Yu et al., 2021; Su et al., 2019; Li et al., 2019b; Yao et al., 2022) exhibit decent performance by correctly selecting the answers in the multiple-choice format. However, a general-purpose region understanding model has yet to emerge. In this paper, by harnessing the powerful large language model (Touvron et al., 2023a; Chiang et al., 2023), GPT4RoI uses a generative approach to handle all these tasks. Users can complete various region-level understanding tasks by freely asking questions.

## 2.4 Using textual coordinates as the grounding token.

We compare the design philosophy with methods using textual coordinates as the grounding token and provide a brief overview of concurrent works, all of which can be found in the appendix.

## 3 Method: GPT4RoI

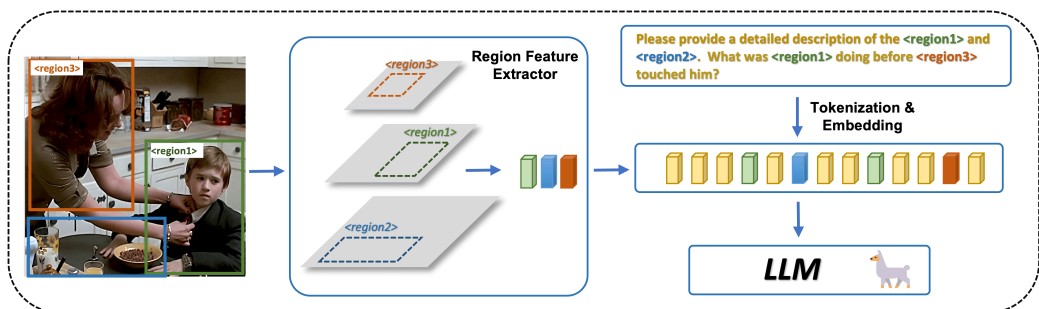

Figure 2: GPT4RoI is an end-to-end vision-language model for processing spatial instructions that contain references to the region-of-interest, such as *<region{i}>*. During tokenization and conversion to embeddings, the embedding of *<region{i}>* in the instruction is replaced with the RoIAlign results from multi-level image features. Subsequently, such an interleaved region feature and language embedding sequence can be sent to a large language model (LLM) for further processing. We also utilize the entire image feature to capture global information and omit it in the figure for brevity. A more detailed framework figure can be found in Figure 5 in the Appendix.

The overall framework of GPT4RoI consists of a vision encoder, a projector for image-level features, a region feature extractor, and a large language model (LLM). Compared to previous works (Zhu et al., 2023; Liu et al., 2023a), GPT4RoI stands out for its ability to convert instructions that include spatial positions into an interleaved sequence of region features and text embeddings, as shown in Figure 2.

## 3.1 Model Architecture

We adopt the ViT-L/14 architecture from CLIP (Radford et al., 2021) as the vision encoder. Following (Liu et al., 2023a), we use the feature map of the penultimate transformer layer as the representation of the entire image, and then map the image feature embedding to the language space using a single linear layer as projector. Finally, we employ the Vicuna (Zheng et al., 2023), an instruction-tuned LLaMA (Touvron et al., 2023a), to perform further processing.

We utilize widely adopted modules in the field of object detection to construct our RoI feature extractor. To ensure a robust feature representation for regions of varying scales, we construct a multi-level image feature pyramid (Lin et al., 2017) by selecting four layers from the CLIP vision

encoder and fusing them with five lightweight scale shuffle modules (Zhang et al., 2023d). These layers are located at the second-to-last, fifth-to-last, eighth-to-last, and eleventh-to-last positions, respectively. Additionally, we incorporate feature coordinates (Liu et al., 2018a; Wang et al., 2020) for each level to address the problem of translation invariance in CNNs. This helps make the model sensitive to absolute position information, such as the description *"girl on left"* in Figure 3. Finally, we use RoIAlign to extract region-level features with an output size of $14 \times 14$ (He et al., 2017), which ensures that sufficient detailed information is preserved. Moreover, all four level features are involved in the RoIAlign operation and fused into a single embedding as the representation of the region of interest (RoI) (Liu et al., 2018b).

## 3.2 TOKENIZATION AND EMBEDDING

To enable users to refer to regions of interest in text inputs, we define a special token *<region{i}>*, which acts as the placeholder that will be replaced by the corresponding region feature after tokenization and embedding. One example is depicted in Figure 2. When a user presents a spatial instruction, *"What was* `<region1>` *doing before* `<region3>` *touched him?"*, the embedding of `<region1>` and `<region3>` are replaced by their corresponding region features. However, this replacement discards the references to different regions. To allows LLM to maintain the original references (*region1, region3*) in the response sequence, the instruction is modified to *"What was region1* `<region1>` *doing before region3* `<region3>` *touched him?"*. Then, LLM can generate a reply like *"The person in region1 was eating breakfast before the person in region3 touched them."*

Regardless of the user instruction, we incorporate a prefix prompt, *"The* `<image>` *provides an overview of the picture."* The `<image>` is a special token that acts as a placeholder, the embedding of which would be replaced by image features of the vision encoder. These features enable LLM to receive comprehensive image information and obtain a holistic understanding of the visual context.

## 3.3 SPATIAL INSTRUCTION TUNING

Our model is trained using a next-token prediction loss (Liu et al., 2023a; Zhu et al., 2023), where the model predicts the next token in a given input text sequence. The training details are in Section A.2 in the Appendix.

We transform annotations into instruction tuning format by creating a question that refers to the mentioned region for each region-text annotation. We partition the available region-text data into two groups, employing each in two distinct training stages. In the first stage, we attempt to align region features with word embeddings in language models using simple region-text pairs that contain color, position, or category information. The second stage is designed to handle more complex concepts, such as actions, relationships, and common sense reasoning. Furthermore, we provide diverse instructions for these datasets to simulate chat-like input in this stage.

**Stage 1: Pre-training** In this stage, we first load the weights of LLaVA (Liu et al., 2023a) after its initial stage of training, which includes a pre-trained vision encoder, a projector for image-level features, and an LLM. We only keep the region feature extractor trainable and aim to align region features with language embedding by collecting short text and bounding box pairs. These pairs are from both normal detection datasets and referring expression detection datasets, which have short expressions. The objective is to enable the model to recognize categories and simple attributes of the region in an image, which are typically represented by a short text annotation (usually within 5 words). Specifically, we utilize COCO (Lin et al., 2014), RefCOCO (Yu et al., 2016), and RefCOCO+ (Yu et al., 2016) datasets in this stage.

As shown in Table 2, for COCO detection data, we first explain the task in the prompt and then convert the annotations to a single-word region caption task. For RefCOCO and RefCOCO+, we also give task definitions first and train the model to generate descriptions containing basic attributes of the region. Only the description of the region (in red color) will be used to calculate the loss.

After this training stage, GPT4RoI can recognize categories, simple attributes, and positions of regions in images, as shown in Figure 3.

**Stage 2: End-to-end fine-tuning** In this stage, we only keep the vision encoder weights fixed and train the region feature extractor, image feature projector, and LLM weights. Our main focus is to

**Object Detection**
In the conversation below, you simply answer the category name based on what you see in the imagery inside a particular region. I will give you only one region each time. Categories containing person, bicycle, car ...
<region1> person
<region2> dog

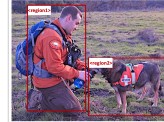

**Referring Expression Comprehension**
I will provide you with only one region containing only one object, although there may be other objects present in the image. It is recommended that you describe the object's relative position with respect to other objects in the image and its basic attributes.
<region1> red shirt girl
<region2> guy in black
<region3> right most person blurred

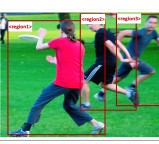

Table 2: The instruction template for Stage 1 training data: For both tasks, we begin by providing a description of the task definition and the expected answer. Then, we concatenate all region-text pairs into a sequence. For detection data, the format is *<region{i}> category_name*. For referring expression comprehension, the format is *<region{i}> description of region*. Only the responses highlighted in red are used to calculate the loss.

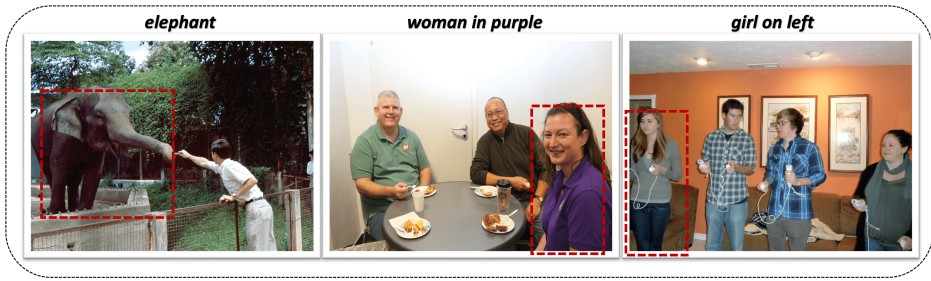

Figure 3: After Stage 1 training, GPT4RoI is capable of identifying the category of the region (elephant), simple attributes such as color (purple), and the position of the region (left).

enhance GPT4RoI's ability to accurately follow user instructions and tackle complex single/multiple region understanding tasks. We tailor specific instructions for different tasks. For single region caption, we construct from Visual Genome (VG) region caption part (Krishna et al., 2017) and RefCOCOg (Mao et al., 2016). For multiple region caption, Flicker30k (Plummer et al., 2015) is converted to a multiple region caption task where the caption should include all visual elements emphasized by bounding boxes. To simulate user instruction, we create 20 questions for each caption task as shown in Table 8 and Table 9. For the region reasoning task, we modify Visual Commonsense Reasoning (VCR) (Zellers et al., 2019a) to meet the input format requirements and make it more similar to human input. The details of this process can be found in Section A.3.

To improve the capability of GPT4RoI for multi-round conversation and generate more human-like responses, we also involve the LLaVA150k (Liu et al., 2023a) visual instruction dataset in this stage. We employ an off-the-shelf LVIS detector (Fang et al., 2023) to extract up to 100 detection boxes per image. These boxes are then concatenated with the user instructions in the format *"<region{i}> may feature a class_name"*. LLaVA150k significantly improves the capability of GPT4RoI for multi-round conversation .

After completing this training stage, GPT4RoI is capable of performing complex region understanding tasks based on user instructions, including region caption and reasoning, as demonstrated in Section 4.

# 4 DEMOSTRATIONS

In this section, we compare the differences between the *visual* instruction tuning model LLaVA (Liu et al., 2023a) and our *spatial* instruction tuning model GPT4RoI. We demonstrate our new interactive approach and highlight its advanced capabilities in understanding multimodality.

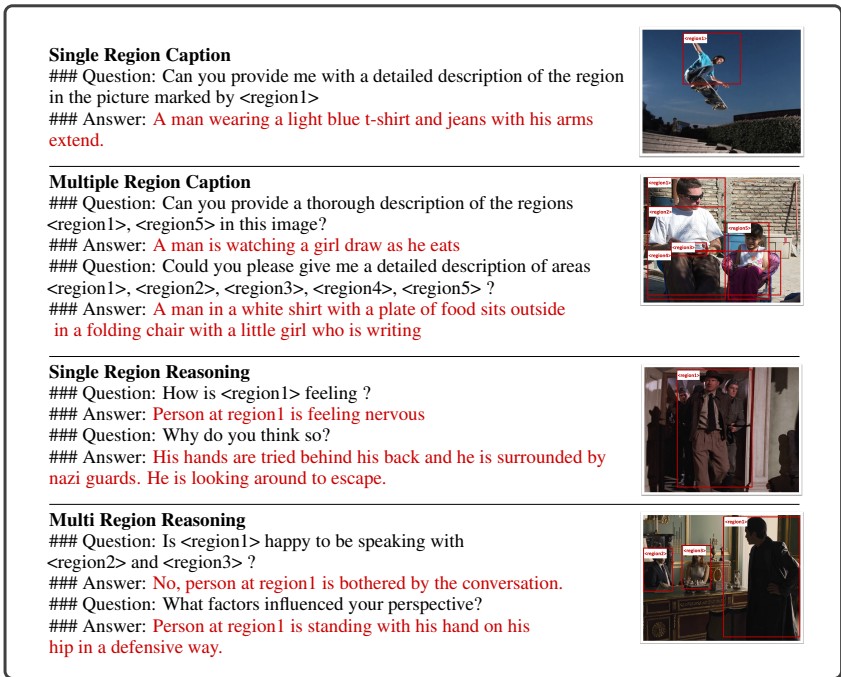

Table 3: Instruction template for Stage 2 training data: During training, we randomly select one question for both single and multiple region caption tasks. For reasoning tasks, we modify the original questions to include a reference for each region so that GPT4RoI can mention them in its response. Only the response in red color and stop string **###** will be used to calculate the loss.

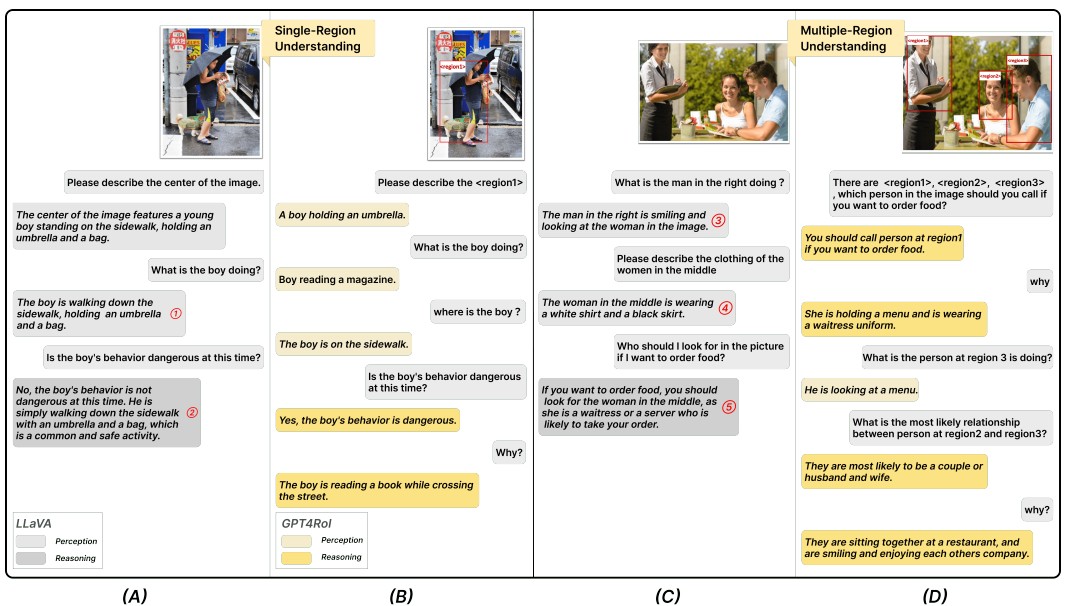

Figure 4: GPT4RoI and LLaVA dialogue performance showcase. Figures A and C demonstrate the dialogue scenarios of LLaVA when referring to a single instance and multiple instances solely using natural language in the conversation. On the other hand, Figures B and D showcase how GPT4RoI utilizes bounding boxes as references to address the same scenarios.

As shown in Figure 4.A, when we try to make LLaVA focus on the center region of the image, it only sees the boy holding an umbrella and a bag, but it misses the book. As a result, LLaVA gives a

wrong answer to the question *"What is the boy doing"* (Figure 4.A.①), and this leads to an incorrect conclusion that *"the boy's behavior is not dangerous"* (Figure 4.A.②).

In comparison, as shown in Figure 4.B, our approach GPT4RoI efficiently recognizes visual details using the given bounding box. This allows it to accurately identify the action of *"reading a magazine."* Furthermore, GPT4RoI demonstrates its reasoning abilities by correctly inferring that the *"boy's behavior is dangerous"*, and giving a reasonable reason that *"the boy is reading a book while crossing the street"*.

When there are multiple instances in the image (as depicted in Figure 4.C), we attempt to refer to the corresponding instances as *"the right"* and *"the middle"*. However, LLaVA provides incorrect information by stating that the right man is *"looking at the women"* (as shown in Figure 4.C.③). Even more concerning, LLaVA overlooks the actual women in the middle and mistakenly associates the women on the left as the reference, resulting in completely inaccurate information (as shown in Figure 4.C.④ & ⑤).

In comparison, as shown in Figure 4.D, GPT4RoI is able to understand the user's requirements, such as identifying the person to call when ordering food, and accurately recognize that the person in region1 fulfills this criterion. Additionally, it correctly recognizes that the person in region3 is *"looking at the menu"*. Importantly, GPT4RoI can also infer relationships between the provided regions based on visual observations. For example, it deduces that the likely relationship between region2 and region3 is that of a *"couple"*, providing a reasonable explanation that they *"are smiling and enjoying each other's company"*.

## 5 QUANTITATIVE RESULTS

To quantitatively evaluate GPT4RoI, we have chosen three representative benchmarks to assess the region understanding capabilities. These benchmarks include the region caption task on Visual Genome (Krishna et al., 2017), the region reasoning task on Visual-7W (Zhu et al., 2016), and Visual Commonsense Reasoning (Zellers et al., 2019a) (VCR). In order to minimize the impact of specific dataset label styles and make evaluation metrics easier to calculate, we fine-tuned GPT4RoI on each benchmark using different task prompts. More details can be found in Section A.2 in the Appendix.

### 5.1 REGION CAPTION

We report the scores of BLEU, METEOR, ROUGE, and CIDEr for both GPT4RoI-7B and GPT4RoI-13B on the validation set of Visual Genome (Krishna et al., 2017). The grounding box in the annotation is combined with the task prompt in Appendix Table 7 to get the response.

| Model | BLEU@4 | METEOR | ROUGE | CIDEr |
|---|---|---|---|---|
| GRiT (Wu et al., 2022) | - | 17.1 | - | 142.0 |
| GPT4RoI-7B | 11.5 | 17.4 | 35.0 | 145.2 |
| GPT4RoI-13B | 11.7 | 17.6 | 35.2 | 146.8 |

Table 4: Comparision of region caption ability on the validation dataset on Visual Genome. All methods employ ground truth bounding boxes and GPT4RoI can outperform previous state-of-the-art specialist GRiT.

The generalist approach GPT4RoI outperforms the previous state-of-the-art specialist model GRiT (Wu et al., 2022) by a significant margin, without any additional techniques or tricks. Additionally, we observe that the performance of GPT4RoI-7B and GPT4RoI-13B is comparable, suggesting that the bottleneck in performance lies in the design of the visual module and the availability of region-text pair data. These areas can be explored further in future work.

### 5.2 VISUAL-7W

Visual-7W (Zhu et al., 2016) is a PointQA dataset that contains a which box setting. Here, the model is required to choose the appropriate box among four options, based on a given description. For example, a question might ask, *"Which is the black machine under the sign?"*. This type of question

not only tests the model's object recognition but also its ability to determine the relationship between objects.

To prevent information leakage, we remove overlapping images with the test set from Visual Genome (Krishna et al., 2017). The results clearly demonstrate that the 13B model outperforms the 7B model by a significant margin. This finding suggests that the reasoning ability heavily relies on the Large Language Model (LLM).

| Model | LSTM-Att (Zhu et al., 2016) | CMNs (Hu et al., 2016) | 12in1 (Lu et al., 2020) | GPT4RoI-7B | GPT4RoI-13B |
|---|---|---|---|---|---|
| Acc(%) | 56.10 | 72.53 | 83.35 | 81.83 | 84.82 |

Table 5: Accuracy on Visual-7W test dataset.

## 5.3 VISUAL COMMONSENSE REASONING

Visual Commonsense Reasoning (VCR) offers a highly demanding scenario that necessitates advanced reasoning abilities, heavily relying on common sense. Given the question(Q), the model's task is not only to select the correct answer(A) but also to select a rationale(R) that explains why the chosen answer is true. We give a more detailed explanation of each metric in our appendix

| Model | Open Source | Parameters | Val Acc.(%) | | | Test Acc.(%) | | |
|---|---|---|---|---|---|---|---|---|
| | | | $Q \rightarrow A$ | $QA \rightarrow R$ | $Q \rightarrow AR$ | $Q \rightarrow A$ | $QA \rightarrow R$ | $Q \rightarrow AR$ |
| ViLBERT (Lu et al., 2019) | Y | 221M | 72.4 | 74.5 | 54.0 | 73.3 | 74.6 | 54.8 |
| Unicoder-VL (Li et al., 2019a) | Y | - | 72.6 | 74.5 | 54.5 | 73.4 | 74.4 | 54.9 |
| VLBERT-L (Su et al., 2019) | Y | 383M | 75.5 | 77.9 | 58.9 | 75.8 | 78.4 | 59.7 |
| UNITER-L (Chen et al., 2020) | Y | 303M | - | - | - | 77.3 | 80.8 | 62.8 |
| ERNIE-ViL-L (Yu et al., 2021) | Y | - | 78.52 | 83.37 | 65.81 | 79.2 | 83.5 | 66.3 |
| MERLOT (Zellers et al., 2021) | Y | 223M | - | - | - | 80.6 | 80.4 | 65.1 |
| VILLA-L (Gan et al., 2020) | Y | - | 78.45 | 82.57 | 65.18 | 78.9 | 82.8 | 65.7 |
| RESERVE-L (Zellers et al., 2022) | Y | 644M | - | - | - | 84.0 | 84.9 | 72.0 |
| VQA-GNN-L (Wang et al., 2022) | Y | 1B+ | - | - | - | 85.2 | 86.6 | 74.0 |
| GPT4RoI-7B | Y | 7B+ | **87.4** | **89.6** | **78.6** | - | - | - |
| VLUA+@Kuaishou | N | - | - | - | - | 84.8 | 87.0 | 74.0 |
| KS-MGSR@KDDI Research and SNAP | N | - | - | - | - | 85.3 | 86.9 | 74.3 |
| SP-VCR@Shopee | N | - | - | - | - | 83.6 | 88.6 | 74.4 |
| HunYuan-VCR@Tencent | N | - | - | - | - | 85.8 | 88.0 | 75.6 |
| Human Performance (Zellers et al., 2019a) | - | - | - | - | - | 91.0 | 93.0 | 85.0 |
| GPT4RoI-13B | Y | 13B+ | - | - | - | **89.4** | **91.0** | **81.6** |

Table 6: Accuracy scores on VCR. GPT4RoI achieves state-of-the-art accuracy among all methods.

GPT4RoI shows significant improvements over the previous methods across all $Q \rightarrow A$, $QA \rightarrow R$, and $Q \rightarrow AR$ tasks. Notably, in the crucial $Q \rightarrow AR$ task, GPT4RoI-13B achieves a performance of 81.6 accuracy, surpassing preceding methods by over 6 points, even outperforming confidential company-level results, which may take advantage of private data. Our totally open-source pipeline can make GPT4RoI a solid baseline. More importantly, this performance is almost reaching human-level performance of 85.0 accuracy, which shows that the multimodal ability of GPT4RoI is promising to be further developed to human intelligence. Furthermore, comparing GPT4RoI to previous methods, particularly observing the size of the language model used, also demonstrates the significant benefits of the Large Language Model (LLM) for visual reasoning tasks.

## 6 CONCLUSIONS

In this paper, we present GPT4RoI, an end-to-end vision-language model that can execute user instructions to achieve region-level image understanding. Our approach employs spatial instruction tuning for the large language model (LLM), where we convert the reference to bounding boxes from user instructions into region features. These region features, along with language embeddings, are combined to create an input sequence for the large language model. By utilizing existing open-source region-text pair datasets, we show that GPT4RoI enhances user interaction by accurately referring to regions and achieves impressive performance in region-level image understanding tasks.

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

# A    APPENDIX

In this appendix, we provide a detailed method architecture figure. We then discuss training-related details, including hyperparameters and instruction templates used in each stage and task. Specifically, we give an introduction for VCR dataset and describe how we utilize the VCR dataset. We also compare the design philosophy with methods using textual coordinates in LLM and provide a brief overview of concurrent works . Finally, we analyze some error cases and propose potential improvements for future exploration.

## A.1    DETAILED ARCHITECTURE

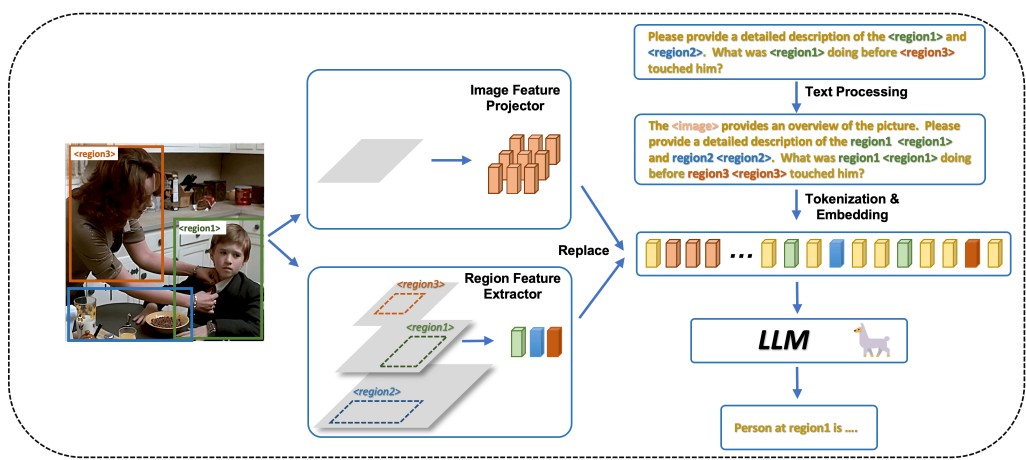

Figure 5: A more detailed framework of GPT4RoI.

Here is a more detailed framework of our approach, GPT4RoI.

1. We preprocess the input text by adding prefixes to retain both image information and pure text references for each region.
2. Next, we tokenize and embed the text. The image feature and region features will replace the placeholders *<image>* and *<region{i}>* respectively.
3. The resulting interleaved sequence of region & image features and language embeddings is then fed into a large language model (LLM) for further processing.

## A.2    TRAINING DETAILS

**Dialogue model** The dialogue model in the demo is trained on 8 GPUs, each with 80G of memory. During the first training stage, a learning rate of 2e-5 is used with a cosine learning schedule. The batch size is 16 for 2 epochs, with a warm-up iteration set to 3000 and a warm-up ratio of 0.003. The weight decay for all modules was set to 0. During the second training stage, the learning rate is reduced to 2e-5 and the model is trained for 1 epoch. To enable end-to-end fine-tuning of the model, which includes a 7B Vicuna, Fully Sharded Data Parallel (FSDP) is enabled in PyTorch to save memory.

**Downstream tasks** We finetune on three datasets with different learning schedules and task prompts (as shown in Table 7). For the region caption task on Visual Genome (Krishna et al., 2017), we perform fine-tuning for 4 epochs with a learning rate of 2e-5. As for Visual-7W (Zhu et al., 2016), we observe that it requires a smaller learning rate of 1e-6 to stabilize the training, which is also trained in 2 epochs. On the Visual Commonsense Reasoning (Zellers et al., 2019a), we fine-tune the model for 1 epoch using a learning rate of 2e-5.

**Instruction of three downstream tasks.** The instructions for three downstream tasks are provided in Table 7.

**Region Caption Task on Visual Genome**
### Question: Can you give a description of the region mentioned by <region>
### Answer: A man wearing a light blue t-shirt and jeans with his arms extended

**Region Reasoning Task on Visual-7W**
### Question: <region1>,<region2>,<region3>,<region4> refers to specific areas within the photo along with their respective identifiers. I need you to answer the question. Questions are multiple-choice; you only need to pick the correct answer from the given options (A), (B), (C), or (D). Which is the black machine under the sign?

### Answer: (A)

**Region Reasoning Task on VCR**
**Q → A**
### Question: <region1>,<region2>,<region3>... refers to specific areas within the photo along with their respective identifiers. I need you to answer the question. Questions are multiple-choice; you only need to pick the correct answer from the given options (A), (B), (C), or (D).

How is 1 feeling ?
(A),1 is feeling amused .
(B),1 is upset and disgusted .
(C),1 is feeling very scared .
(D),1 is feeling uncomfortable with 3

### Answer: (C)
**QA → R**
### Question: <region1>,<region2>,<region3>... refers to specific areas within the photo along with their respective identifiers. I give you a question and its answer, I need you to provide a rationale explaining why the answer is right. Both questions are multiple-choice; you only need to pick the correct answer from the given options (A), (B), (C), or (D).

"How is 1 feeling ?" The answer is "1 is feeling very scared." What's the rationale for this decision?
(A),1's face has wide eyes and an open mouth .
(B),When people have their mouth back like that and their eyebrows lowered they are usually disgusted by what they see .
(C),3,2,1 are seated at a dining table where food would be served to them . people unaccustomed to odd or foreign dishes may make disgusted looks at the thought of eating it .
(D),1's expression is twisted in disgust .

### Answer: (A)

Table 7: Task prompt of three downstream tasks.

**Instruction of Single-Region Caption** The instructions for single-region caption are provided in Table 8. We randomly select one as the question in training.

**Instruction of Multi-Region Caption** The instructions for multi-region caption are provided in Table 9. We randomly select one as the question in training.

## A.3    VCR

**Introduction to the VCR Dataset**    The Visual Commonsense Reasoning(VCR) dataset (Zellers et al., 2019b), comprises 290,000 multiple-choice questions obtained from 110,000 movie scenes. Each image in the dataset is annotated with a question that requires common-sense reasoning, along with its corresponding answer and the explanation for the answer. VCR is a particularly challenging dataset for comprehension and reasoning. It has gained attention from several well-known organizations, who have submitted their solutions on the leaderboard. The dataset's distinctive challenge is that a model not only needs to answer complex visual questions but also provide a rationale for why its answer is correct. The VCR task consists of two sub-tasks: Question Answering (Q→A) and Answer Justification (QA→R). In the Q→A setup, a model is given a question and must

select the correct answer from four choices. In the QA->R setup, a model is provided with a question and the correct answer, and it needs to justify the answer by selecting the most appropriate rationale from four choices. The performance of models is evaluated using the Q→AR metric, where accuracy is measured as the percentage of correctly answered questions along with the correct rationale.

**Preprocess of VCR**    To construct a sequence of questions, we convert the explanation to a follow-up question and format them into a two-round conversation. Table 10 shows an example of the follow-up question that asks for the reasoning behind the answer.

The VCR dataset is valued for its diverse question-answer pairs that require referencing from prior question-answers to perform reasoning. Therefore, it's crucial to assign a reference to each region in the dataset. We accomplish this by starting each conversation with a reference to all regions, e.g., *There are <region1>, <region2>... in the image.* This approach explicitly references every region, avoiding confusion in future analyses. Additionally, we substitute the corresponding *<region{i}>* in the answer with *category_name at region{i}* to ensure a plain text output sequence.

## A.4 TEXTUAL COORDINATES AS THE GROUNDING TOKEN

The key distinction lies in whether to incorporate the detection function into the LLM. For the method that uses textual coordinates as the grounding token, they have to solve the following challenge:

Aligning a large number of position tokens with their corresponding positions in the image by training on a large set of datasets. But this is actually a simple rule that can be naturally implemented with the operation in detection architectures.

Modeling geometric properties can be challenging. For example, if the ground truth box is $< x_1 = 0, y_1 = 0, x_2 = 5, y_2 = 5 >$, a predicted box of $< x_1 = 1, y_1 = 1, x_2 = 4, y_2 = 4 >$ would be considered a better result than $< x_1 = 1, y_1 = 1, x_2 = 8, y_2 = 8 >$. because it has a higher overlap with the ground truth. However, incorporating this geometric property into the next token prediction task using cross-entropy loss can be challenging. On the other hand, utilizing traditional loss functions such as L1 or IoU loss can naturally handle this geometric constraint.

Dense to Sparse  (Ren et al., 2015; Zhang et al., 2023d) is a crucial design for detection performance, but embedding such an idea into the sequential form of LLM is challenging. We provide two pieces of evidence to support our argument

1. The performance of pix2seq (Chen et al., 2021; 2022), which utilizes object365 (Shao et al., 2019) pretrain, falls significantly behind the corresponding specialist  (Zhang et al., 2022; Li et al., 2023a).

2. Even with scaled-up data and parameters, GPT4V still faces challenges in object counting (Yang et al., 2023b). However, this is a trivial task for detection methods.

Another approach is to use an external detector to find the potential region of interest, whereas LLM only focuses on analyzing the corresponding region of interest. This is the motivation of GPT4RoI. It requires much less data and allows for quick adaptation to specific domain problems with the corresponding detector. However, the drawback is that the framework may appear less elegant and it assumes input contains all regions of interest that need to be analyzed.

Both approaches have their advantages and disadvantages, and academic research in both directions is thriving (including concurrent works or follow-ups on GPT4RoI). For the first approach, relevant references include (Zhao et al., 2023; Chen et al., 2023b), while for the second approach, there are (Anonymous, 2023; Chen et al., 2023a) besides GPT4RoI. Additionally, there has been research that explores a fusion of the two approaches, as shown in references (You et al., 2023; Rasheed et al., 2023; Zhang et al., 2023b).

## A.5 FAILURE CASE ANALYSIS

Due to limited data and instructions, GPT4RoI may fail in several landmark scenarios. We have conducted a thorough analysis and look forward to improving these limitations in future versions.

**Instruction obfuscation** As shown in Figure 6.(a), our multiple-region reasoning capability mainly relies on VCR, where we often use sentences that declare <region1>, <region2>, etc. at the beginning of the question. However, when users adopt the less common sentence structure to refer to regions, it

can often be confused with region captions that have the highest proportion in the dataset. As shown in Figure 6.(b), because our data and instructions are mainly generated by rules, our training data does not include content with the "respectively" instruction in multi-region scenarios. This can be resolved by adding specific instructions. In future versions, we aim to develop more diverse instructions, while ensuring data balance.

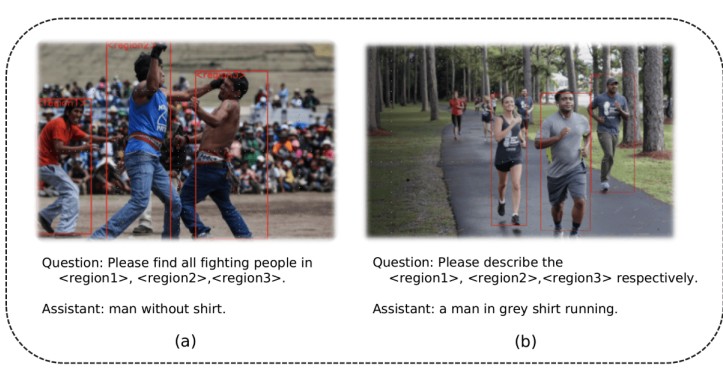

Figure 6: **GPT4RoI on instruction obfuscation.**

**Misidentification of fine-grained information within in region** Although GPT4RoI has improved the fine-grained perception ability of images compared to image-level vision language models, the limited amount of region-level data results in insufficient fine-grained alignment within regions. For example, in Figure 7.(a), the model incorrectly identifies the color of the helmet, and in Figure 7.(b), it misidentifies the object in the girl's hand. Both cases generate the corresponding answers based on the most prominent feature within the region. Using semi-supervised methods to create more region-level data may address this issue.

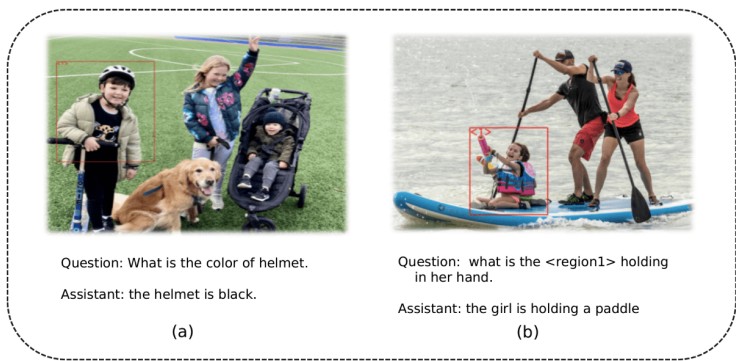

Figure 7: **GPT4RoI on Misidentification of fine-grained information.**

## A.6 DISCUSSION

In our exploration, we find GPT4RoI produces failure cases as shown in Section. A.5. To further improve the performance, we identify the following potential directions:

- Model architecture. We find that $224 \times 224$ input image resolution struggles with understanding smaller regions. However, if we switch to a larger resolution, we must consider the potential burden on inference speed from global attention ViT architecture, while the more efficient CNN architecture or sliding window attention has no available pre-trained large-scale vision encoder like CLIP ViT-H/14.
- More region-text pair data. The amount of available region-text pairs is notably smaller than that of image-text pairs, which makes it challenging to sufficiently align region-level features

with language models. To tackle this issue, we may try to generate region-level pseudo labels by leveraging off-the-shelf detectors to generate bounding boxes for image-text data.

- Region-level instructions. Although we have generated instructions for each task from existing open-source datasets, users in practical applications may ask various questions about an arbitrary number of regions, and the existing data may not contain satisfactory answers. To tackle this issue, we suggest generating a new batch of spatial instructions through manual labeling or by leveraging ChatGPT or GPT4.

- Interaction mode. Currently, GPT4RoI only supports natural language and bounding box interaction. Incorporating more open-ended interaction modes such as point, scribble, or image-based search could further improve the user interaction experience.

---

1. Can you provide me with a detailed description of the region in the picture marked by <region1>?
2. I'm curious about the region represented by <region1> in the picture. Could you describe it in detail?
3. What can you tell me about the region indicated by <region1> in the image?
4. I'd like to know more about the area in the photo labeled <region1>. Can you give me a detailed description?
5. Could you describe the region shown as <region1> in the picture in great detail?
6. What details can you give me about the region outlined by <region1> in the photo?
7. Please provide me with a comprehensive description of the region marked with <region1> in the image.
8. Can you give me a detailed account of the region labeled as <region1> in the picture?
9. I'm interested in learning more about the region represented by <region1> in the photo. Can you describe it in detail?
10. What is the region outlined by <region1> in the picture like? Could you give me a detailed description, please?
11. Can you provide me with a detailed description of the region in the picture marked by <region1>, please?
12. I'm curious about the region represented by <region1> in the picture. Could you describe it in detail, please?
13. What can you tell me about the region indicated by <region1> in the image, exactly?
14. I'd like to know more about the area in the photo labeled <region1>, please. Can you give me a detailed description?
15. Could you describe the region shown as <region1> in the picture in great detail, please?
16. What details can you give me about the region outlined by <region1> in the photo, please?
17. Please provide me with a comprehensive description of the region marked with <region1> in the image, please.
18. Can you give me a detailed account of the region labeled as <region1> in the picture, please?
19. I'm interested in learning more about the region represented by <region1> in the photo. Can you describe it in detail, please?
20. What is the region outlined by <region1> in the picture like, please? Could you give me a detailed description?

Table 8: A list of instructions for single-region caption.

1. Could you please give me a detailed description of these areas [<region1>, <region2>, ...]?

2. Can you provide a thorough description of the regions [<region1>, <region2>, ...] in this image?

3. Please describe in detail the contents of the boxed areas [<region1>, <region2>, ...].

4. Could you give a comprehensive explanation of what can be found within [<region1>, <region2>, ...] in the picture?

5. Could you give me an elaborate explanation of the [<region1>, <region2>, ...] regions in this picture?

6. Can you provide a comprehensive description of the areas identified by [<region1>, <region2>, ...] in this photo?

7. Help me understand the specific locations labeled [<region1>, <region2>, ...] in this picture in detail, please.

8. What is the detailed information about the areas marked by [<region1>, <region2>, ...] in this image?

9. Could you provide me with a detailed analysis of the regions designated [<region1>, <region2>, ...] in this photo?

10. What are the specific features of the areas marked [<region1>, <region2>, ...] in this picture that you can describe in detail?

11. Could you elaborate on the regions identified by [<region1>, <region2>, ...] in this image?

12. What can you tell me about the areas labeled [<region1>, <region2>, ...] in this picture?

13. Can you provide a thorough analysis of the specific locations designated [<region1>, <region2>, ...] in this photo?

14. I am interested in learning more about the regions marked [<region1>, <region2>, ...] in this image. Can you provide me with more information?

15. Could you please provide a detailed description of the areas identified by [<region1>, <region2>, ...] in this photo?

16. What is the significance of the regions labeled [<region1>, <region2>, ...] in this picture?

17. I would like to know more about the specific locations designated [<region1>, <region2>, ...] in this image. Can you provide me with more information?

18. Can you provide a detailed breakdown of the regions marked [<region1>, <region2>, ...] in this photo?

19. What specific features can you tell me about the areas identified by [<region1>, <region2>, ...] in this picture?

20. Could you please provide a comprehensive explanation of the locations labeled [<region1>, <region2>, ...] in this image?

Table 9: A list of instructions for multiple-region caption.

1. Why?
2. What's the rationale for your decision
3. What led you to that conclusion?
4. What's the reasoning behind your opinion?
5. Can you explain the basis for your thinking?
6. What factors influenced your perspective?
7. How did you arrive at that perspective?
8. What evidence supports your viewpoint?
9. What's the logic behind your argument?
10. Can you provide some context for your opinion?
11. What's the basis for your assertion?
12. What experiences have shaped your perspective?
13. What assumptions underlie your reasoning?
14. What's the foundation of your assertion?
15. What's the source of your reasoning?
16. What's the motivation behind your decision?
17. What's the impetus for your belief?
18. What's the driving force behind your conclusion?
19. What's your reasoning?
20. What makes you say that?
21. What's the story behind that?
22. What's your thought process?
23. What's the deal with that?
24. What's the logic behind it?
25. What's the real deal here?
26. What's the reason behind it?
27. What's the rationale for your opinion?
28. What's the background to that?
29. What's the evidence that supports your view?
30. What's the explanation for that?

Table 10: A list of instructions for the second round chat in VCR.

