# OpenReview forum: "GPT4RoI: Instruction Tuning Large Language Model on Region-of-Interest"
_ICLR.cc/2024/Conference — Submitted to ICLR 2024_

### Official Review · Reviewer_yj7F · 2023-10-14

**Soundness:** 4 excellent
**Presentation:** 4 excellent
**Contribution:** 3 good
**Rating:** 8
**Confidence:** 3

**Summary:**

This submission proposes a finer-grained visual instruction tuning to endow a pretrained large language model with the capabilities to “see.” Compared to the literature, this work is different in that it has region-level image understanding and flexibility in combining multiple ROI’s and reasoning between them. It initializes the model from pretrained on vision and language tasks, respectively, and then uses a two-step instruction tuning pipeline to build multiple visual reasoning capabilities into it.

**Strengths:**

In general, the motivation, technique, presentation, and results look all solid to me. The anonymously released codebase looks usable, which, if true, would serve as a well-compiled dataset and starting point for later multi-modality research.

**Weaknesses:**

I have to say I haven’t followed multimodality research for some time, so my judgment on the novelty of this work (compared to the literature) could be rusty. This might be an important aspect to consider for this work, but my score is assuming this work is novel.

On presentation, a flowchart of the two-stage training process might be useful to readers. Indicating which datasets are used in which fashion at which stage, with the purpose of bringing about which kind of capabilities.

**Questions:**

Some of my random thoughts, potentially useful to the authors:
(1)	How about visual question-answering datasets?
(2)	Is it possible to design some synthetic tasks to train the model, perhaps inserted between the two stages you have right now, to smooth up the training process?
(3)	Is it possible to let the LM output coordinates somehow so to interact with users by referring to certain regions in the picture?

---

> ### Author Response · Authors · 2023-11-12
> **Response to Reviewer yj7F**
>
> Dear Reviewer yj7F,
>
> Thank you for your positive comments on our work. Our open-source code is complete and reproducible, and it has been widely adopted by the community. As the first work to utilize LLM to achieve general-purpose region understanding, we believe that we can serve as a solid baseline in this field.
>
> Below are our responses to  your suggestions:
>
> (1). A flowchart of the two-stage training process
>
> In the current presentation format, it does appear that the two stages with multiple datasets are somewhat confusing. Introducing a flowchart could indeed be a viable improvement, and we will strive to enhance the presentation format in future versions accordingly. Thank you for your valuable suggestion.
>
> (2). How about visual question-answering datasets?
>
> When aiming to enhance performance in image-level VQA tasks, a practical approach for GPT4RoI could involve integrating an off-shelter detector and combining region of interest features with image features to represent an image. Previous research [1] has demonstrated the effectiveness of this technique in image caption task. Given the specific focus of our work on region-understanding tasks, we will further investigate this approach in our new research.
>
> (3). Design some synthetic tasks
>
> Utilizing region-text pair data and ChatGPT, building region-level multi-round understanding and reasoning tasks is a practical way to improve the dialogue ability of GPT4RoI. A related study [2] has conducted similar work following our research.
>
> (4). LM output coordinates
>
> This suggestion is highly valuable. Currently, GPT4RoI can only mention the input RoI and requires users or detectors to provide all the region of interest. Two remarkable follow-up works [3][4]  of GPT4RoI utilize a significant amount of visual grounding data, allowing LLM to output new regions. This enables the model to have an enhanced interactive experience. We will continuously improve the GPT4RoI to consolidate these two functions in the future.
>
> [1] Faster and Better Image Captioning Transformer Using Dual Visual Features
>
> [2] Position-Enhanced Visual Instruction Tuning for Multimodal Large Language Models
>
> [3] Ferret: Refer and Ground Anything Anywhere at Any Granularity
>
> [4] GLaMM : Pixel Grounding Large Multimodal Model

---

> > ### Comment · Reviewer_yj7F · 2023-11-12
> > **Reply to reply**
> >
> > Cool! Thank you for answering my questions. Good luck!

---

### Official Review · Reviewer_fWBC · 2023-10-30

**Soundness:** 3 good
**Presentation:** 3 good
**Contribution:** 3 good
**Rating:** 6
**Confidence:** 4

**Summary:**

The paper introduced the region-of-interest instruction with vision language models. By doing this, it proposed spatial instruction, combining language and the reference to region-of-interest I to an interleave sequence, enabling accurate region referring and enhancing user interaction. By spatial instruction tuning LLM with massive region-text datasets, the model can follow user instructions to solve diverse region understanding tasks, such as region caption and reasoning. The results show that the model outperforms the previous state-of-the-art approach on a wide range of region understanding benchmarks.

**Strengths:**

- The paper proposed a novel method with vision language tasks. It provides a detailed methodology for this.
- A comprehensive discussion of experiments and results, where the figures are in good quality and readability.
- The benchmark methods are of good quality.

**Weaknesses:**

After Rebuttal: Upon re-reviewing the manuscript and checking other fellow reviewers' comments, I have identified several major concerns that I previously overlooked.
- My biggest concern: the model is evaluated on the Visual Genome, Visual-7W, and VCR datasets. However, if I understand correctly, the model has been pre-trained on the Visual Genome and VCR datasets. I am therefore concerned that the model's strong performance on these two datasets is a case of overfitting, especially given that LLMs have a high capacity for overfitting. Can you provide more experimental results on different tasks and datasets (that are not used for pre-training)?
- The motivation for fusing different levels of visual and textual features is not clearly explained. More analysis on why certain levels are chosen would be helpful. Why does spatial instruction tuning yield better performance? Because more training data is used? Meanwhile, can you please provide statistics on the amount of training data used in the baselines in the tables? It could show whether the performance improvement of the model is due to the proposed method or more amount of training data.
- There are no ablation studies showing the contribution of individual components like visual encoders, textual encoders etc.
- (Minor concerns) How sensitive is the model to the choice vision encoders? What is the training and inference time compared to baseline models? Does the model exhibit biases like relying more on textual or visual features for certain question types?

Given the potential for overfitting, I find it necessary to adjust my score to 6. However, I would still vote to accept the paper. Many thanks! (btw, I apologize for the delayed response)

Original: For region caption, the paper only compares the proposal to one model GRiT, which is limited.

**Questions:**

Please see above for details:
Can you interpret the performance of different tasks (ie Q->A, QA-> R, Q->AR) in Table 6?

---

> ### Author Response · Authors · 2023-11-12
> **Response to Reviewer fWBC**
>
> Dear Reviewer fWBC,
>
> Thank you for your appreciation of our work, particularly for recognizing the novelty and experimental results. Below are my responses to some of your concerns.
>
> (1). Why only compare to GRiT
>
> To assess the region's comprehension capabilities, we rely on ground truth bounding boxes or detection model outputs as input. Consequently, access to the method source code is necessary for conducting the evaluation. **GRiT is the top-1 performance model among all open-sourced projects in the region-caption field**, and serves as a representative baseline in recent studies when specifically evaluating region-caption abilities. We will continuously track the latest advancements in the region caption field and update our results accordingly. We appreciate your valuable feedback and suggestions.
>
> (2). Q->A, QA-> R, Q->AR performance on VCR
>
> The VCR dataset is a particularly difficult dataset for comprehension and reasoning. It has gained attention from several well-known companies, who have submitted their solutions on the leaderboard. The dataset's distinctive challenge is that a model not only needs to answer complex visual questions but also provide a rationale for why its answer is correct.
>
> The VCR task consists of two subtasks: Question Answering (Q->A) and Answer Justification (QA->R). In the Q->A setup, a model is given a question and must select the correct answer from four choices. In the QA->R setup, a model is provided with a question and the correct answer, and it needs to justify the answer by selecting the most appropriate rationale from four choices. The performance of models is evaluated using the Q->AR metric, where accuracy is measured as the percentage of correctly answered questions along with the correct rationale.

---

### Official Review · Reviewer_MYqZ · 2023-10-31

**Soundness:** 2 fair
**Presentation:** 3 good
**Contribution:** 2 fair
**Rating:** 3
**Confidence:** 3

**Summary:**

This work introduces a new approach called spatial instruction tuning, which aims to enhance the fine-grained multimodal understanding of vision-language models. The proposed model, GPT4RoI, incorporates references to regions-of-interest (RoI) in instructions by replacing them with RoI features and interleaving them with language embeddings. By training on region-text pair datasets, GPT4RoI enables interactive and conversational experiences, allowing users to interact with the model through both language and drawing bounding boxes. GPT4RoI achieves remarkable results on the Visual Commonsense Reasoning (VCR) dataset.

**Strengths:**

-	Fine-grained multimodal understanding: GPT4RoI enables region-level alignment and understanding by incorporating references to RoIs in instructions, allowing for more detailed analysis and reasoning.
-	Interactive user experience: Users can interact with GPT4RoI through both language input and drawing bounding boxes.
-	GPT4RoI achieves remarkable accuracy on the VCR dataset, surpassing existing models by a significant margin.

**Weaknesses:**

-	Expanding from image-level to region-level instruction tuning seems like a natural progression, and the approach is straightforward without providing a fresh perspective. Some other papers also explore the region-level large language models [1] but lack the performance comparison.
-	It appears that while this paper utilized more datasets for training, the improvement in results is relatively marginal, as shown in Table 5.
-	This work lacks a comparison of parameters. The current models seem to be quite large, especially large language models. A comparison should be conducted at the same parameter level, e.g., Table 6.

[1] ChatSpot: Bootstrapping Multimodal LLMs via Precise Referring Instruction Tuning.

**Questions:**

See Weaknesses

-	Comparisons to other works on the same parameter level.

---

> ### Author Response · Authors · 2023-11-12
> **Response to Reviewer MYqZ: [1/2]**
>
> We would like to express our sincere gratitude to Reviewer MYqZ for their valuable time and effort in reviewing our work. We are grateful for your recognition of the interactive user experience and the impact of our approach on VCR tasks. Below, we address each identified weakness.
>
> (1). Regarding question 1, about the novelty and comparison with ChatSpot:
>
> - **Expanding from image-level to region-level instruction tuning is a fundamental progression in multimodal large language models**
>
>     The emergence of ChatGPT(LLM) is well-recognized as a new era of deep learning, just like AlexNet in 2012. At that time, how to expand AlexNet used for image classification task to object-level recognition tasks is a big challenge for the whole community, until the proposal of R-CNN in 2014. Although R-CNN is a quite simple design, I believe no one would say that it is just a natural progression from AlexNet to R-CNN. Similarly,  our proposed GPT4RoI is motivated to expand the LLM’s visual instruction tuning from image-level to region-level and serves a fundamental role in the related area.
>
>     A large portion of academic researchers pursue sophisticated designs and even agree that complexity is equal to innovation, this is definitely wrong! No matter Occam’s razor or practical applications is telling ***a basic rule: the simpler, the better***. GPT4RoI incorporates region references in instruction and developing the pioneering model for general-purpose region understanding, we definitely believe it is novel and provides a fresh perspective for the region understanding field.
>
> - **Concurrent Independent Work: impossible to compare when submitting the manuscript**
>
>     It is important to note that GPT4RoI **predates** ChatSpot by **about two weeks** and serves as a concurrent research project. Furthermore, the unavailability of ChatSpot's code poses challenges in conducting a comprehensive comparative analysis. Based on their report, ChatSpot addresses the reference problem by incorporating position tokens, which may require more region data to realign the position token with the image position compared to GPT4RoI. Additionally, it is worth mentioning that ChatSpot concatenates region position at the end of the instruction, while GPT4RoI formulates interleaved instructions. This difference makes it challenging for ChatSpot to handle multiple-region understanding tasks like VCR. As a verification, ChatSpot has not presented any demo for multi-region understanding.

---

> ### Author Response · Authors · 2023-11-12
> **Response to Reviewer MYqZ: [2/2]**
>
> (2). Regarding question 2, utilizing more datasets for training, the improvement in results is relatively marginal, as shown in Table 5.
>
> - The baseline **12-in-1[1]** in Table.5 utilizes 12 pre-training datasets, clearly indicating the use of more data compared to our method.
> - Table.5 (visual 7W) is just one of the benchmarks for reasoning ability. On the other hand, the VCR dataset (Table.6), which has gained considerable attention in recent years and attracted submissions from many companies, is more popular and challenging. We have demonstrated our strong performance on this dataset with significantly less training data, achieving an improvement of **6 points** compared to the previous state-of-the-art. Therefore, we believe that labeling our method's improvement in results as relatively marginal is one-sided.
>
> (3). Lacks a comparison of parameters in Table.6.
>
> The table below presents the rank, and accuracy of Q-AR, along with the parameters and datasets of various solutions, from the most recent **[VCR leaderboard](https://visualcommonsense.com/leaderboard/)** update on November 11, 2023. We retained the top 6 methods and 2 methods that have reports.
>
> | model | rank | Q-AR(%) | Parameters | Datasets | Open Source |
> | --- | --- | --- | --- | --- | --- |
> | GPT4RoI | 1 | 81.6 | 13B+ | 7 open-source region data | Y |
> | HunYuan | 2 | 75.6 | confidential | confidential | N |
> | Shopee | 3 | 74.4 | confidential(ensemble of 4 models) | confidential | N |
> | KDDI Research and SNAP | 4 | 74.3 | confidential | confidential | N |
> | Kuaishou | 5 | 74.0 | confidential | confidential | N |
> | VQA-GNN + MerlotReserve-Large | 6 | 74.0 | 1B+ (ensemble of 2 models) | YT-Temporal-1B | N |
> | MerlotReserve-Large | 11 | 71.5 | 644M | YT-Temporal-1B | Y |
> | UNIMO+ERNIE | 12 | 71.4 | confidential（ensemble of 7 models），each maybe 355 M | BookWiki, OpenWebText, COCO, Visual Genome, Conceptual Captions, SBU Captions | - |
>
> Finding fair comparisons among solutions on the VCR leaderboard can be challenging due to the confidentiality of most submissions. Even in open-source projects, fair comparisons are often overlooked due to the use of different technical approaches, data, training methods, and even model ensembles. In this diverse landscape, our method, GPT4RoI, stands out by utilizing fully open-source region-level data and surpassing all previous approaches by a significant margin (over 6 points) using just a single model. We believe that our method, as the pioneering usage of LLM for the VCR task with a completely open-source pipeline, establishes a robust new baseline.
>
> [1] *12*-in-*1*: *Multi*-*Task Vision and Language Representation Learning*

---

> ### Author Response · Authors · 2023-11-15
> **Does our rebuttal address your concerns?**
>
> Dear reviewer MYqZ,
>
> Thank you for reviewing our work and engaging in the rebuttal process to enhance the quality of the paper.
>
> Have our rebuttal adequately addressed your concerns? If you still have any issues with our rebuttal or if there are any new concerns, we are more than willing to continue the discussion with you.

---

> ### Author Response · Authors · 2023-11-22
> **Last discussion request before the discussion period ends**
>
> Dear Reviewer MYqZ,
>
> I hope this message finds you well. The author-reviewer discussion period is drawing to a close, and we look forward to receiving your response at this last moment.
>
> As it is in response, there may be some misunderstanding for you of the results in Table 5. We have also taken your suggestions and created a more comprehensive Table 6 in the main manuscript which contains more details about previous methods. However, due to the confidential nature of the previously state-of-the-art methods, we were unable to obtain detailed information. Additionally, the open-source solutions are far behind our approach. Therefore, we truly believe that our approach - being ***fully open source*** and achieving a ***landmark performance improvement of 6 points*** and ***ranking 1st on the leaderboard*** - can become a solid baseline.  We have also addressed each of your other questions in detail and sincerely hope that these responses have alleviated any concerns you may have had.
>
> Given the limited time remaining, I would be grateful if you could take some time to review our responses. If you feel that our responses have addressed your concerns, we would be most appreciative if you could consider changing your initial rating. If you still have any remaining concerns, we are happy to continue discussing them with you during this final window.
>
> Thank you for your time and consideration.

---

> ### Comment · Reviewer_MYqZ · 2023-11-22
> **Response to the Authors**
>
> Thanks for the authors’ responses. I have the following three questions (No need to experiment):
>
> - 12-in-1 utilizes 12 pre-training datasets but what is the amount of data samples compared to this work? Simply providing the magnitude comparison of the data samples is OK.
> - As stated in my initial question, is it possible to perform comparisons with other works on the same parameter level? As shown in Table 6, VQA-GNN-L uses 1B+ parameters but this work uses 7B+ parameters. Can this 7B parameter be reduced to a smaller size?
> - Also in Table 6, can the results of GPT4RoI-7B on the test set be listed for comparison with VQA-GNN-L?

---

> ### Author Response · Authors · 2023-11-22
> **Response to the MYqZ**
>
> Dear reviewer MYqZ,
>
> Thank you for your response to our rebuttal.
>
> Q1: Magnitude comparison of the data samples
>
> |Method | # data samples |
> |---------| ------------|
> | 12-in-1| 4.4 million|
> |GPT4RoI (ours) | 0.43 million |
>
> We use only a subset of the data that is used by 12-in-1, which is ten times smaller (0.43 million versus 4.4 million).
>
> Q2: Smaller GPT4RoI
>
> Our approach is pioneering in ***utilizing a Large Language Model (LLM)*** for creating a versatile region understanding model.  The term 'Large' in LLM reflects its substantial parameter size.  We've employed the 7B version, the smallest in the community. So we regretfully inform you that it's currently beyond our capabilities for your suggested experiment. However, we are confident that our work has already made a significant impact by ***demonstrating the potential of LLMs to enhance visual reasoning ability***.
>
> Q3: Test Results of GPT4RoI-7B
>
> Firstly, the VCR test set results need to be manually obtained by emailing the author, but the author has joined OpenAI and seems to be very busy. Even after we sent multiple emails, it still took about 10 days to receive a response with the 13B test results. We are sending an email of GPT4RoI-7B now.
>
> However, another fact may already resolve your concerns. You will notice from Table 6 and all previous studies that the ***results on the test set are about 0.4+ higher than those on the validation set (I haven't seen an exception)***. Therefore, the result of 7B should be 5 points higher than VQA-GNN + MerlotReserve-Large. We believe this is a ***landmark improvement*** because ***previous methods have been blocked at around 75\% for almost a year***. We believe that our method is meaningful enough in this field.
>
> Thank you very much for your prompt response. If you have any other questions, please reply to us promptly because time seems to be very urgent. We hope to resolve your issues as soon as possible.

---

> ### Author Response · Authors · 2023-11-23
> **There are less than 5 hours remaining for our discussion**
>
> Dear Reviewer MYqZ,
>
> There are less than 5 hours remaining for our discussion, and we sincerely hope that we have adequately addressed your concerns.
>
> We kindly request your thorough consideration of the notable contributions made by our work. We have developed the ***first LLM-based region understanding model that can follow user instructions***, and our quantitative results have demonstrated its outstanding superiority. Notably, we achieved a ***groundbreaking 6-point improvement***, ***surpassing the performance bottleneck that has persisted in VCR for over a year***.
>
> Since ChatGPT showcased the effectiveness of large models, there has been a shift in focus towards ***designing the entire system***. Various datasets and pre-trained models are being utilized to address problems, ***rather than making minor enhancements to a mature pipeline through meticulous ablations***. GPT4RoI represents a novel paradigm utilizing LLM, distinct from previous methods. We understand your desire for fair comparisons of parameters with traditional methods. However, we must acknowledge that it is indeed challenging for the first method based on LLM. We genuinely appreciate your understanding and consideration of these factors and give your final rating.

---

### Official Review · Reviewer_kN3t · 2023-11-01

**Soundness:** 3 good
**Presentation:** 3 good
**Contribution:** 2 fair
**Rating:** 5
**Confidence:** 4

**Summary:**

This paper proposes to enhance current large multimodal models by injecting regional awareness.
Authors leverage the public available regional data as the instruction data.

**Strengths:**

1. Both qualitative and quantitative results demonstrate that now the model can have a sense of location.
 2. The presentation is clear.
3. The figures are easy to read.

**Weaknesses:**

1. Which part of the model design leads to positional awareness is unclear. Authors have " five lightweight scale shuffle modules", "ROI Align", "add feature coordinates (Liu et al., 2018) for each level (positional embedding)", "extract region-level features with the output size of 14×14", which part really makes the model work? There is no ablation study.
2. Finetuning on a specific dataset can lead to the case that the model forgets all other knowledge. For example, fine-tuning on the multichoice dataset will lead to the case that model can not speak out natural languages whatever you ask.
3. From the qualitative examples, seems like the model can only produce short descriptions, which may not be suitable when answering with long context. I think the author's model may overfit to such datasets with short captions.
4. When converting LLaVa instruct dataset into the format with bounding box, I cast doubt on how accurate it is. If many instances appear in the same image, is it ok to attach so many detection results beforehand? Besides, the LLaVa instruct dataset is known to include hallucination
5. How do authors claim on other methods which use textual coordinate as the grounding token? In this way, they can  not only use bbox as input, but also use them in output.

**Questions:**

1. What is the specific vision model you used? Official CLIP model does not have ViT-H at all.
2.  For each region, it is compressed into one token? It is enough?

---

> ### Author Response · Authors · 2023-11-13
> **Response to Reviewer kN3t: [1/3]**
>
> Dear Reviewer kN3t,
>
> Thank you for the appreciation of our work, especially the recognition of our model's ability in spatial referencing and the positive feedback on our paper presentation.
>
> (1). For Question 1, the core design makes GPT4RoI  positional awareness
>
> The core design of GPT4RoI  incorporates a region feature extractor and replaces the ROI reference in the instruction with the extracted region feature. The implementation of the region feature extractor can be flexible and varied, as mentioned in our introduction, such as using deformable attention or RoI Align operation. We opted for RoI Align in our study due to its simplicity and widespread familiarity among researchers.
> The region feature extractor is a fundamental component in object detection and instance segmentation. It can be considered that every detail and parameter of this component has been thoroughly ablated.
>
>  We provide a detailed explanation of the origin and significance of each sub-module for better understanding.
>   1. The Scale Shuffle module is an algorithm that we have chosen from the multi-level feature fusing modules [1, 2, 3]. These modules are typically utilized to provide robust feature representation for regions of varying scales. They play a crucial role as an essential component in enhancing the performance of all instance recognition tasks.
>   2. RoI Align is an operation that is employed to extract region features by utilizing the coordinates of a bounding box. It serves as the fundamental module in R-CNN detectors preceding DETR-style ones.
>   3. To address the problem of translation invariance in CNNs when absolute position information is required in your task, a commonly employed technique is to augment feature maps with feature coordinates[4, 5, 6].
>   4. When aiming to extract more detailed information within a region, we usually increase the RoI Align resolution. such as  from (7x7) in object detection[7] to (14x14) in instance segmentation[8].
>
>
> (2). For Question 2, fine-tuning on a specific benchmark to do the evaluation
>
> As pioneers in the field of region-understanding general-purpose models, there is currently no specific benchmark available for evaluating region-level dialogue in this context. Creating such an evaluation benchmark falls outside the scope of our research. Therefore, we have chosen to conduct our testing on traditional region understanding tasks, specifically multiple-choice questions. It is common practice in this field to fine-tune models on multiple-choice question tasks to evaluate performance. For instance, LLaVA [9] is fine-tuned for image-level understanding tasks, such as science QA [10].
>
> [1] Scale-Equalizing Pyramid Convolution for Object Detection
>
> [2] Unifying Object Detection Heads with Attentions
>
> [3] Dense Distinct Query for End-to-End Object Detection
>
> [4] An intriguing failing of convolutional neural networks and the coordconv solution
>
> [5] SOLO: Segmenting Objects by Locations
>
> [6] SOLOv2: Dynamic and Fast Instance Segmentation
>
> [7] Faster R-CNN: Towards Real-Time Object Detection with Region Proposal Networks
>
> [8] Mask R-CNN for Object Detection and Segmentation
>
> [9] Visual Instruction Tuning
>
> [10] Learn to Explain: Multimodal Reasoning via Thought Chains for Science Question Answering

---

> > ### Author Response · Authors · 2023-11-13
> > **Response to Reviewer kN3t: [2/3]**
> >
> > (3). For Question 3, answering with long context
> >
> > Not answering meaningful long context is a common issue for current large vision language models, since all open-source data only includes brief text descriptions. To mitigate this problem, LLaVa[1] and MiniGPT4[2] utilize ChatGPT/GPT4 to refine these short captions. However, we believe that this approach has limited significance. This is because the additional description is inferred by the LLM based on suspicion and often lacks accuracy or informative value. To achieve accurate and meaningful long context, the only way is to involve human participation in a data annotation pipeline to establish large datasets. We consider this to be beyond the scope of this paper.
> >
> >
> > (4). For Question 4, usage of LLaVA instruct dataset
> >
> > As mentioned in the introduction, our approach involves detection results to leverage LLaVA data in order to enhance the multi-round conversation ability. This is because our other constructed datasets consist of only single or two-round conversations. To this end,  we do not need to specifically consider whether the detection results strictly correspond to the text in the LLaVA data. From the demonstration results, it can be observed that GPT4ROI has acquired the ability to engage in multi-round conversations.
> >
> > [1] Visual Instruction Tuning
> >
> > [2] MiniGPT-4: Enhancing Vision-Language Understanding with Advanced Large Language Models

---

> ### Author Response · Authors · 2023-11-13
> **Response to Reviewer kN3t: [3/3]**
>
> (5). For Question 5, About using textual coordinates as the grounding token.
>
> This is an extremely valuable question that we have dedicated a lot of thought to. We are delighted to share these perspectives with you.
>
> The key distinction lies in whether to incorporate the detection function into the LLM. For the method that uses textual coordinates as the grounding token, They have to solve the following challenge
>
> - Aligning a large number of position tokens with their corresponding positions in the image by training on a large set of datasets. But this is actually a simple rule that can be naturally implemented with the operation in detection architectures.
>
> - Modeling geometric properties can be challenging. For example,  if the ground truth box is  $<x_1=0, y_1=0, x_2=5, y_2=5>$, a predicted box of $<x_1=1, y_1=1, x_2=4, y_2=4>$ would be considered a better result than $<x_1=1, y_1=1, x_2=8, y_2=8>$. because it has a higher overlap with the ground truth. However, incorporating this geometric property into the next token prediction task using cross-entropy loss can be challenging. On the other hand, utilizing traditional loss functions such as L1 or IoU loss can naturally handle this geometric constraint.
> - Dense to Sparse [1][4] is a crucial design for detection performance, but embedding such an idea into the sequential form of LLM is challenging.
>
> We provide two pieces of evidence to support our argument
>
> - The performance of pix2seq[2][3], which utilizes object365 pretrain, falls significantly behind the corresponding specialist [4][5].
> - Even with scaled-up data and parameters, GPT4V still faces challenges in object counting[6]. However, this is a trivial task for detection methods.
>
> Another approach is to use an external detector to find the potential region of interest, whereas LLM only focuses on analyzing the corresponding region of interest. This is the motivation of GPT4RoI. It requires much less data and allows for quick adaptation to specific domain problems with the corresponding detector. However, the drawback is that the framework may appear less elegant and it assumes input contains all region of interest that need to be analyzed.
>
> Both approaches have their advantages and disadvantages, and academic research in both directions is thriving (including concurrent works or follow-ups on GPT4RoI). For the first approach, relevant references include [7, 8], while for the second approach, there are [9, 10] besides GPT4RoI. Additionally, there has been research that explores a fusion of the two approaches, as shown in references [11, 12, 13].
>
> ***It is still too early to make a definitive judgment on which approach has absolute superiority at this time.***
>
>
> (6). For Question 6, about vision model architecture
>
> We apologize for the mistake. We used the ViT-L/14 transformer architecture and will fix this in the next version. Thank you for pointing it out.
>
> (7). For Question 7, about number of tokens for each region
>
> Our experiments found that increasing the number of tokens does not make a difference in these open-source datasets and corresponding benchmarks.
>
>
> [1] Faster R-CNN: Towards Real-Time Object Detection with Region Proposal Networks
>
> [2] Pix2seq: A Language Modeling Framework for Object Detection
>
> [3] A Unified Sequence Interface for Vision Tasks
>
> [4] DINO: DETR with Improved DeNoising Anchor Boxes for End-to-End Object Detection
>
> [5] Mask DINO: Towards A Unified Transformer-based Framework for Object Detection and Segmentation
>
> [6] The Dawn of LMMs: Preliminary Explorations with GPT-4V(ision)
>
> [7] ChatSpot: Bootstrapping Multimodal LLMs via Precise Referring Instruction Tuning
>
> [8] Shikra: Unleashing Multimodal LLM's Referential Dialogue Magic
>
> [9] Ins-DetCLIP: Aligning Detection Model to Follow Human-Language Instruction
>
> [10] Position-Enhanced Visual Instruction Tuning for Multimodal Large Language Models
>
> [11] Ferret: Refer and Ground Anything Anywhere at Any Granularity
>
> [12] GLaMM : Pixel Grounding Large Multimodal Model
>
> [13] NExT-Chat: An LMM for Chat, Detection and Segmentation

---

> ### Author Response · Authors · 2023-11-16
> **Looking forward to your reply**
>
> Dear reviewer kN3t,
>
> We sincerely appreciate your review of our work and your valuable contribution to improving the quality of the paper.
>
> Have we effectively addressed your concerns in our rebuttal? If you still have any issues or new concerns, please feel free to let us know so we can continue the discussion.

---

> ### Author Response · Authors · 2023-11-22
> **Discussion request before the discussion period ends**
>
> Dear Reviewer kN3t,
>
> The author-reviewer discussion period is drawing to a close. We have provided responses to each of your questions above and made some improvements to the main manuscript following your suggestion. We have added an in-depth discussion regarding the methods that utilize textual coordinates as grounding tokens and enhanced the clarity of the RoI extractor section in the main manuscript.
>
> Given the limited time remaining, I would be grateful if you could take some time to review our responses. If you feel that our responses have addressed your concerns, we would be most appreciative if you could consider changing your initial rating. If you still have any remaining concerns, we are happy to continue discussing them with you during this final window.
>
> Thank you for your time and consideration.

---

> ### Author Response · Authors · 2023-11-23
> **Last Discussion request before the discussion period ends**
>
> Dear Reviewer kN3t,
>
> There are less than 5 hours remaining for our discussion, and we sincerely hope that we have adequately addressed your concerns.
>
> We have made every effort to revise the article and address your concerns. At this final moment, we also hope that you can take note of the highlights of our work, specifically the ***first LLM-based region understanding model that can follow user instructions***. Additionally, we would like to draw your attention to its significant progress on VCR. Our model achieved a remarkable performance ***improvement of 6 points***, ***ranking 1st on the leaderboard***. It overcame the long-standing performance bottleneck in VCR, which ***persisted for over a year***. We kindly ask you to consider our rebuttal and make your final rating based on a comprehensive evaluation.
>
> Thank you for your time and consideration.

---

> > ### Comment · Reviewer_kN3t · 2023-11-23
> >
> > I appreciate authors' response to my questions.
> > I am still curious Is CLIP ROI feature really the key to localizing objects? As early works like RegionCLIP indicate, we actually need to train a region-aware model so that ROI feature can correspond to the feature of a certain region. That being said, directly cropping the CLIP feature will not lead to precise correspondence.
> >
> > I am just wondering in this case, will "ROI" actually makes the proposed methods work. That being said, if we remove the positional embedding, remove the "shuffle", remove the learned FPN parameters, will the pure ROI feature get good performance on tasks such as Visual7W and RefCOCOg?

---

> ### Author Response · Authors · 2023-11-23
> **Response to Reviewer kN3t**
>
> This is an insightful question. However, due to the limited time of ***only two hours remaining***, we will add these results in future versions.
>
> But We believe that the ***absence of these results will not undermine any of our claimed contributions in the article***.
>
> We want to emphasize again that our ***main contribution*** is the ***first general-purpose region understanding model based on LLM***.  The modules you mentioned are constructed based on our experience in the detection field. As a pioneering work, ***our model's performance is already satisfactory***. We didn't provide detailed ablation of these sub-components in the initial version because we believe it is not the contribution of our work. As we mentioned before, these reference object components can be implemented in various forms.
>
>
> As it is the final moment, we kindly request the reviewer to consider ***the overall system and its remarkable performance***. If there are additional specific questions, please feel free to ask, and we will address them in future versions of the paper.

---

### Author Response · Authors · 2023-11-21
**A Summary of Paper Updates**

We sincerely value the constructive feedback given by the reviewers, as it has greatly helped improve our work. Taking their suggestions into account, we have made substantial updates to the paper(marked in blue).

Section 2.4 & Appendix A.4:

Discussing methods using textual coordinates as a grounding token. (Suggestion from Reviewer kN3t)

Section 3.1:

Adding more details about each component of our RoI Extractor to make it clear for readers without object detection background. (Suggestion from Reviewer kN3t)

Section 5.3 & Sub A.4:

As the ***Rank 1*** method of [VCR Leaderboard](https://visualcommonsense.com/leaderboard/), we provide more detailed information about previous methods, such as parameters, and include corresponding discussions. (Suggestion from MYqZ )

Appendix A.3:

More details about the metric in the VCR dataset.  (Suggestion from fWBC)

---

### Meta-Review · Area_Chair_jUAa · 2023-12-10

**Metareview:**

This paper presents a method to improve vision-language referring tasks by integrating region information into a multimodal LLM (mLLM) . The idea is simple, before sending to LLM, the reference in text is replaced by region-of-interests features and interleaved with text token embeddings as a sequence.

Pros (from reviewers):
1. One of the first work to address region level referring and reasoning task in multimodal LLM.
2. Strong results on VCR task.

Cons (from reviewers):
1. The idea of using region feature to intersect with text has been studied in prior work (non autoregressive models).
2. More experiments / evaluation are needed to demonstrate the generalizability of the model.

**Justification For Why Not Higher Score:**

See discussion in the `Additional Comments On Reviewer Discussion`

**Justification For Why Not Lower Score:**

N/A

---

### Decision · Program_Chairs · 2024-01-16

Reject